# Point-of-Care COVID-19 Antigen Testing in Exposed German Healthcare Workers—A Cost Model

**DOI:** 10.3390/ijerph182010767

**Published:** 2021-10-14

**Authors:** Roland Diel, Norbert Hittel, Albert Nienhaus

**Affiliations:** 1Institute for Epidemiology, University Medical Hospital Schleswig-Holstein, 24015 Kiel, Germany; 2Lung Clinic Grosshansdorf, Airway Disease Center North (ARCN), German Center for Lung Research (DZL), 22927 Großhansdorf, Germany; 3Institution for Statutory Accident Insurance and Prevention in the Health and Welfare Services (BGW), 22089 Hamburg, Germany; a.nienhaus@uke.de; 4Otsuka Novel Products GmbH, 80636 München, Germany; NHittel@otsuka-onpg.com; 5Institute for Health Service Research in Dermatology and Nursing, University Medical Center Hamburg-Eppendorf, 20246 Hamburg, Germany

**Keywords:** costs, point-of-care, antigen testing, real-time reverse transcriptase polymerase chain reaction (RT-PCR), SARS-CoV-2, COVID-19

## Abstract

Background: Hospital staffing shortages are again (mid-year 2021) becoming a significant problem as the number of positive COVID-19 cases continues to increase worldwide. Objective: To assess the costs of sending HCW into quarantine (Scenario 1) from the hospital’s and the taxpayer’s perspective versus the costs arising from implementing point-of-care COVID-19 antigen testing (POCT) for those staff members who, despite learning that they have been exposed to hospital patients later found to be infected with COVID-19, continue to report to work (Scenario 2). Methods: A mathematical model was built to calculate the costs of a sample-and-stay strategy for exposed healthcare workers (HCW) in Germany by utilizing a high-quality antigen fluorescent immunoassay (FIA), compared to the costs of quarantine. Direct costs and wage costs were evaluated from the hospital as well as from the taxpayer perspective assuming a SARS-CoV-2 infection prevalence of 10%. Results: Serial POCT testing of exposed HCW in Germany (Scenario 2) who do not go into quarantine but continue to work during a post-exposure period of 14 days at their working place raises costs of EUR 289 (±20%: EUR 231 to EUR 346, rounded) per HCW at the expense of the employing hospital while the extra-costs to the taxpayer per exposed HCW are limited to EUR 16 (±20%: EUR 13 to EUR 19). In contrast, sending HCW into quarantine (Scenario 1) would result in costs of EUR 111 (±20%: EUR 89 to EUR 133) per exposed HCW for the hospital but EUR 2235 (±20%: EUR 1744 to EUR 2727) per HCW at the expense of the taxpayer. Conclusions: Monitoring exposed HCW who continued working by sequential POCT may considerably reduce costs from the perspective of the taxpayer and help mitigate personnel shortages in hospitals during pandemic COVID-19 waves.

## 1. Introduction

The severe acute respiratory syndrome COVID-19, caused by coronavirus 2 (SARS-CoV-2), first appeared in December 2019 in Wuhan, China, with an accumulation of pneumonia and has since spread across the globe [1]. Clinical features of the disease, known as COVID-19, include fever, headache, and cough, but more severe symptoms such as shortness of breath and respiratory failure have also been reported [2]. As of 9 August 2021, around 204 million cases and more than 4.3 million deaths have been registered in nearly 200 countries worldwide [3].

The rapid escalation of the situation caused the World Health Organization to declare a pandemic on 11 March 2020 [4]. Since then, the continued human-to-human transmission of SARS-CoV-2 creates tremendous challenges for healthcare systems and public health laboratories. SARS-CoV-2 infections in healthcare workers (HCW), i.e., all hospital staff members who are in direct contact with a patient, follow those of the general population, but HCW are often at higher risk for SARS-CoV-2 exposure through undetected COVID-19 patients. As the newly named Delta variant is currently spreading to European countries in the coming months, the expected surge (“fourth wave”) in COVID-19 patients might put pressure on hospital staffing. Although in the current situation in Germany, few actual staff shortages have yet to be reported, exposing more healthcare personnel to the highly contagious virus may result in forcing them to stay home. As the number of qualified nurses and doctors for inpatient care in Germany is already insufficient, notions of hiring short-term replacements for quarantined HCW are hardly realistic [5].

The current reference test used to establish SARS-CoV-2 infection worldwide is the real-time reverse transcriptase polymerase chain reaction (RT-PCR). However, in Germany, currently 71.5% of all hospitals have eliminated their in-house laboratories [6]. To ensure the correct diagnosis, respiratory specimens of COVID-19 patients in hospitals or their contact persons suspected to be infected with COVID-19 must usually be sent to external labs for centralized RT-PCR testing, thus resulting in a time-lag of one day before the report of the test result becomes available.

In contrast, lateral flow assay (LFA) SARS-CoV-2 antigen tests, which are based on lateral flow technology that uses monoclonal antibodies, can be performed at point of care, i.e., in the hospital itself, provide results within 15–30 min and are inexpensive. Numerous SARS-CoV-2 POC antigen tests are currently available, whose use offers the potential for rapid identification of those individuals who are not only infected, but infectious and are therefore at greatest risk of spreading the infection [7]. Although, due to methodological reasons, the detection limit for SARS-CoV-2 RNA material out of clinical samples tested by RT-PCR is always lower than the detection limit for SARS-CoV-2 antigen, antigen testing is meanwhile a fixed component of the German National test regulation in hospitals where COVID infections occur [8]. In such hospitals, HCW have the right to be tested without concrete grounds once a week in order to exclude infectivity from undetected SARS-CoV-2 infection. As HCW, if they become contact persons of a COVID-19 patient, are sent home to quarantine for at least 14 days in the same way as contact persons in the community, the question arises whether POCT should not be extended in a periodical manner during the 14-day period for those HCW who do not want to be sent home into quarantine but stay to work in the hospital on a voluntary basis. This concept may be of importance because the problem of personnel shortage must urgently be mitigated in the course of the COVID-19 pandemic. On the other hand, secondary SARS-CoV-2 infections in exposed HCW must also be detected in time in order to prevent further spread of the disease. According to the current CDC recommendations [9], outbreak control depends largely on the frequency of antigen testing and the speed of reporting and is only marginally enhanced by improving the sensitivity of the test. Serial antigen testing every 3 days will almost always identify SARS-CoV-2 during early stages of infection, and thus significantly reduce disease transmission [10,11].

In our model, we separately calculate the economic consequences of a sample-and-stay strategy, which uses periodic POCT at least once every 48 h together with the wearing of a FFP2 mask compared to the costs of sending these HCW into quarantine during a period of 14-days. We did not include the healthcare costs arising from symptomatically infected HCW requiring treatment; such costs have to be covered by the German Public Statutory insurance and thus lie beyond the perspective of the hospital as employer and that of the taxpayer. Also, additional costs arising through quarantine caused by food suppliers or other services could not be considered.

## 2. Materials and Methods

### 2.1. Strategy for Managing Exposed HCW

With respect to exposed HCW, two scenarios are considered: In the first one, as usually performed in Germany, the HCW are sampled for a RT-PCR test immediately at the hospital in order not to overlook a very early infection during the gap of 1–2 days when the increasing viral load may still be below the antigen tests’ limit of detection. In case of a negative result, the HCW will then go home into quarantine for 14 days. If the HCW does not have symptoms, a final antigen test must be done after 14 days, the negative result of which ends the quarantine [12].

In the second scenario, by way of derogation from the first one [13], exposed HCW are subjected to POCT in the hospital every second day and sent to work with strict adherence to the hygiene rules, use of personal protective equipment (PPE) and careful self-observation with regard to the possible appearance of symptoms as long as they remain asymptomatic and test negative.

Relying on recently published figures, we assumed a sensitivity of 80% and a specificity of 98.9% for POC antigen-testing in symptomatically infected COVID-19 subjects [14]. Due to methodological particularities, sensitivity of POCT in asymptomatic persons is generally lower than that in symptomatic patients. In a new Cochrane meta-analysis [7], the pooled sensitivity in asymptomatic participants was 58.1%.

The cost parameters used in our calculation are all derived from fixed or pre-weighted costs. Our calculation does not take into consideration outcomes from clinical trials or different size samples. Therefore, we always use mean costs. Confidence intervals are not provided, as their application to this non-probabilistic model would have been inappropriate. However, we performed a sensitivity analysis by varying all costs by ±20% of the base-case value according to international practice.

### 2.2. Incubation Period and Prevalence of Infections in Exposed HCW

The incubation period for COVID-19 has a median time of 5–6 days from exposure to onset of symptoms and is thought to extend to 14 days [15] thereafter. Consequently, the probability of becoming infected within the frame of 14-days is not evenly distributed over time. According to McAloon’s meta-analysis [16], the incubation period distribution may be modelled assuming a lognormal distribution with pooled µ = 1.63 (95% CI 1.51 to 1.75), sigma = 0.50 (95% CI 0.46 to 0.55) (95% CIs) and corresponding mean = 5.8 (95% CI 5.0 to 6.7) days. We built a mathematical model to derive the probability of infections each second day and to determine the number of true and false positive symptomatic infected persons as well as the true and false negative POCT results at each date of POCT (see Appendix A).

Persons fully vaccinated against COVID-19 are exempt from quarantine measures after exposure to a confirmed COVID-19 case [17] as they have only a low risk of acquiring a “breakthrough infection”. However, as transmission may occur despite vaccination, transmission of COVID-19 to HCW depends not only on vaccination coverage, but also on the use of PPE in the investigated working units. In their cohort studies, Rivett et al. [18] found 30 infected HCW out of 1032 (3%) in the UK and Reusken et al. [19], 45 infected HCW (4.1%) out of 1097 in the Netherlands. Vimercati et al. [20] found that 10 out of 76 HCWs (2.13% of exposed workers) were not PPE-protected and reported a COVID-19 prevalence of 13.16% (10/76) in this high-risk group. In our model, we take a path between the extremes and assume that 10% of exposed HCW become freshly infected by an unknown COVID-19 patient.

### 2.3. Costs of Testing and Outcomes

Absence due to quarantine, which is usually ordered and checked by the locally responsible health department according to para 28 Section 1 of the German Law on Protection against Infection (IfSG), is fully reimbursed to the hospital employing the HCW by the responsible authority, i.e., by the taxpayer, upon request as long as the HCW does not develop disease—in which case the HCW are put on sick leave and the hospital has to absorb the costs of the sick leave days from that date. That means that during the 14-day period of quarantine, the taxpayer must bear the wage costs as long as no symptoms due to SARS-CoV-2 infections appear.

For the second scenario, our model is, as an example, parametrized by data on sensitivity and specificity of the Sofia^®^ SARS Antigen FIA, the final result of which is available in 15 min [21]. Consequently, false positive and false negative POCT results have to be considered (see Appendix A). Given the high specificity, but only moderate sensitivity of that antigen test (98.9% and 80.0% [14]), additional RT-PCR testing of the patient´s samples is always required in contact persons, in this case HCW, who become symptomatic even when the antigen test is scored negative. As RT-PCR testing in an external laboratory, where the infected persons´ samples must be sent in addition, ideally has both a sensitivity and specificity approaching 100%; thus false negative and positive POCT results can be corrected. For the purpose of simplification, we assume that symptoms in HCW during the 14-day quarantine are indeed due to COVID-19. On the other hand, according to the current German guidelines [22], positive POCT results occurring in our serial testing scenario must generally be confirmed by RT-PCR. The reason is that approval of antigen tests by the German Paul-Ehrlich-Institut requires only that specificity exceed 97%. The costs of RT-PCR testing incurred by the hospital are reimbursed according to the German Hospital Finance Act (Krankenhausfinanzierungsgesetz, KHG) [23]. Accordingly, RT-PCR testing does not appear to be a cost factor in our model.

According to the most recent evidence [24], 80% of the infections experienced by the HCW would be symptomatic. The remaining 20% of asymptomatic SARS-CoV-2 infections are only revealed by a surprise positive result in the POCT given to conclude the 14-day quarantine period.

If employees decide to continue working, wear FFP2 masks and undergo an antigen test every other day at the workplace, the hospital as employer has not only to pay all wage costs for its HCW who continue working and those who are symptomatically infected, but bears also the costs of the serial testing, i.e., for performing the seven antigen tests that are to be done during the 14-day incubation time. When HCW test positive but remain asymptomatic, they will be sent into isolation by the responsible public health department, and all wages, again, are covered by the taxpayer. In 2019, the gross annual earning of a hospital employee was EUR 60.663 [25]. Accordingly, divided by 365 days, wage costs per day, irrespective of whether they have to be covered by the taxpayer or by the employer, are, on average, EUR 167.58. The pathways of the 2 scenarios are briefly shown in Figure 1 and Figure 2.

Following a recent publication [14], the costs to the system of one single antigen test in Germany are assumed to be EUR 12 and the time for receiving the result of a POCT is specified with 15 min. It is reasonable to calculate the same time period spent on pre-analytics before running the test and properly documenting the results done by either a physician, laboratory worker or colleague at the ward to perform the POCT. The mean costs can be derived by extrapolating the average gross annual earning of a hospital employee (EUR 60.663). Basically, in accordance with Section 3 I of the German Working Hours Act (ArbZG), a standard working time of eight hours per working day applies. On average, there are 255 working days in 2021, equivalent to 2040 h [26]. Accordingly, the mean costs of one minute are EUR 0.496. Thirty minutes spent for preanalytics of the Sofia antigen test and waiting for its result amount to EUR 14.88. Accordingly, the total cost per POCT are EUR 26.88 (EUR 12 plus EUR 14.88). All costs are reported in 2021 Euros (EUR) and were not discounted because of the only short 14-day period of interest. Table 1 shows the cost parameters used in our model.

## 3. Results

Ten percent of the exposed HCW who are assumed to be infected with SARS-CoV-2 express different infection rates on day 2, 4, 6, 8, 10, 12 and 14 of the incubation period (finally summing up to the total of 10%) according to the log normal distribution described above (see Table 2 and the Appendix A for details).

Of those who are infected, 80% become symptomatic, but on different days, and will be put on sick leave. From those dates, the hospital, as employer, has to pay the sick leave days, summing up to EUR 110.64 per exposed HCW within the 14-day period (Table 3).

The days off work before manifestation of the infection and all asymptomatic infections which will only be revealed by the final PCR-test when the incubation period has already passed, are also paid by the taxpayer. Consequently, the total cost to be paid by the taxpayer per quarantined worker is EUR 2346.12 minus EUR 110.64, i.e., EUR 2235.48.

In the second scenario, where exposed HCW continue to work, the EUR 110.64 at the expense of the employing hospital must be considered for those infected HCW who report their symptoms. Furthermore, performing up to seven antigen tests in those infected HCW who continue to work amounts to EUR 178.02 at the expense of the hospital (Table 4) summing up to an amount of EUR 288.66. At the expense of the taxpayer costs of EUR 16.04 arise for those HCW whose asymptomatic infection is revealed by POC testing, given the antigen´s test documented sensitivity of only 58.1% in infected, but asymptomatic persons. However, asymptomatically infected workers are not sick and will be sent into isolation by the responsible Public Health Department.

We assume that HCW who present COVID-like symptoms but are negative by POCT, as well as HCW who test positive by POCT, will be immediately re-tested by RT-PCR and that any SARS-CoV-2 infections will be detected and/or confirmed without further delay. Thus, in the second scenario, only 0.84% asymptomatic but infected HCW remain undetected and are not placed in quarantine.

All in all, when comparing the cost of the two scenarios, the net cost at the expense of the taxpayer is EUR 2235.48 (Scenario 1) minus the EUR 16.04 (Scenario 2) that would be incurred if the second scenario came into effect, i.e., EUR 2219.44 per exposed HCW. As the sick-leave costs to the hospital as employer is the same in both scenarios, the net cost to the hospital, following the second strategy, is limited to the costs of performing the antigen tests (EUR 178.02 per exposed HCW). Reducing the wages at the expense of the taxpayer by 20% in sensitivity analysis while simultaneously increasing the costs of the employing hospital by the same percentages in Scenario 1 would result in costs of EUR 1744.13 and EUR 133.77, respectively. Increasing by 20% the costs of POCT and wages at the expense of the hospital in Scenario 2 whilst reducing the taxpayer’s costs also by 20% increases the costs of the hospital to an amount of EUR 346.39 per HCW sent to work and diminishes the taxpayer´s costs to EUR 12.83. Accordingly, even under such “worst case” conditions from the standpoint of the employer, the net cost to the taxpayer remains considerable with EUR 1731.3 (EUR 1744.13 [Scenario 1] minus EUR 12.83 [Scenario 2]) per exposed HCW, whereas the net cost of the hospital rises to EUR 231.62.

## 4. Discussion

Newer real-time POC tests, such as the Sofia^®^ SARS Antigen FIA, which can claim specificity of nearly 99%, come close to laboratory RT-PCR testing in their ability to very rapidly and reliably exclude the presence in a patient of transmissible COVID-19. In the meantime, asymptomatic staff testing by POC antigen tests has been a core component of some national infection control testing strategies, such as those in the UK, the US and Germany [27,28,29]. Beyond this, using serial POCT to monitor exposed HCW in hospitals would be a novel public health concept in which decisions to send contact persons into quarantine are based on infection screening rather than a categorical containment imperative. Of note, in a recent report of Gabler and coworkers [30], it could be demonstrated that even in the period of increasing vaccination, rapid antigen testing had the largest effect on reducing the number of SARS-CoV-2 infections in Germany. Frequent large-scale rapid testing should remain part of strategies to contain COVID-19.

Due to the potentially increasing pressure on hospitals to develop strategies to mitigate healthcare personnel staffing shortages, the CDC [31]—and in the meantime also the German RKI [32]—is considering allowing HCW with higher-risk exposures to work during the risky 14-day post-exposure period by reducing the length of the quarantine period. According to interim guidance, quarantine may be discontinued after day 7 if no symptoms are reported during daily monitoring and if the result of testing a diagnostic specimen within 48 h before the time of planned quarantine discontinuation (e.g., in anticipation of testing delays) is negative. Our proposal is intended to avoid any quarantine, even a shortened one, but in return recommends intensive serial testing during the whole presumed incubation period. This is in line with current evidence: Smith et al. found a sensitivity of 82.4% success rate in detecting infected individuals at any time during the 14-day infection period by nasal testing every other day with a high-quality antigen test while testing with a nasal RT-PCR used with the same frequency showed only a slightly higher sensitivity of 88.2% [33].

The infection prevalence in exposed HCW under the strict security standards now in place in German hospitals can generally be considered low, with concerns mostly centered around casual attitudes towards personal protection held by a minority in a given hospital ward. Hence one may assume limited exposure risk, under which a high negative predictive value of POCT in such contacts can be expected. We assumed a moderate test sensitivity of 80% for POC antigen-testing for the 80%-symptomatically infected COVID-19 subjects and a 58.1% sensitivity for the remaining 20% infected persons in our model. As symptomatic patients will always be retested with RT-PCR even if the POCT is negative, no infections of symptomatic patients remain unrecognized. Of note, the number of false-negative results in asymptomatically infected HCW infections during the 14-day incubation period is low (0.84 persons out of 100 HCW). Given the fact that wearing FFP2 masks reduces the risk of contagion by 67% [34], the risk of these subjects infecting other colleagues or patients by HCW left undetected by antigen testing can be considered tolerable (2.8 persons out of 1000 exposed HCW).

As the working wages for contact persons of COVID-19 patients in Germany who must remain in quarantine for 14 days are paid by the taxpayer, a total of EUR 2235 would have been transferred to the hospital as employer per exposed HCW. In contrast, serial POCT testing at their workplace (every other day during the 14-day post-exposure period) of exposed HCW who don´t go into quarantine but who volunteer to continue coming to work under a regulatory derogation, results in extra costs at the expense of the employing hospital of only EUR 178 per HCW. Even reducing all costs of the taxpayer by 20% in sensitivity analysis and simultaneously increasing the hospital’s costs by the same percentage shall not change this discrepancy. Although it might seem counterintuitive for a hospital to not accept pay-off, the latter approach avoids a possibly threatening thinning out of the workforce, and thus can be considered a win-win situation.

Our model has several limitations: A general limitation of our study that must be kept in mind when interpreting its results is that, as always, a single-center economic model cannot depict the true local SARS-CoV-2 infection prevalence among exposed HCW. Furthermore, we do not know how many HCW would in fact follow the offer to return to work due to missing data. Finally, the probabilities of becoming infected on different days during the quarantine period could only be derived as point estimators from McAloon´s log-normal distribution. Since that model cannot be varied, a more comprehensive sensitivity analysis demonstrating the robustness of our findings was not possible. Therefore, to validate our estimates, prospective cost studies are required.

## 5. Conclusions

High-quality POC antigen tests, used for serial testing of healthcare workers exposed towards COVID-19 patients in their hospital, can lessen the burden of quarantine on hospital staff. They allow asymptomatic HCW to continue working (or to return to work earlier) on a voluntary basis, while wearing FFP2 masks and maintaining careful self-observation for signs of developing symptoms, and thus help to avoid unnecessary shortage of urgently needed employees. As such, POCT can also reduce costs from the taxpayer perspective and allow resources to be allocated for other precautions. Prospective clinical studies should be undertaken to further evaluate its economic advantages in the immediate future.

## Figures and Tables

**Figure 1 ijerph-18-10767-f001:**
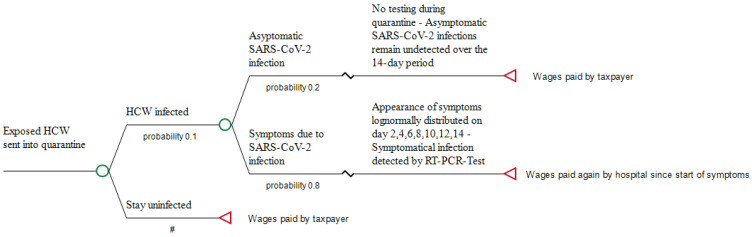
Pathway of sending exposed HCW into quarantine (Scenario 1). #: Complementary probability (all probabilities of chance node’s branches to sum to 1.0).

**Figure 2 ijerph-18-10767-f002:**
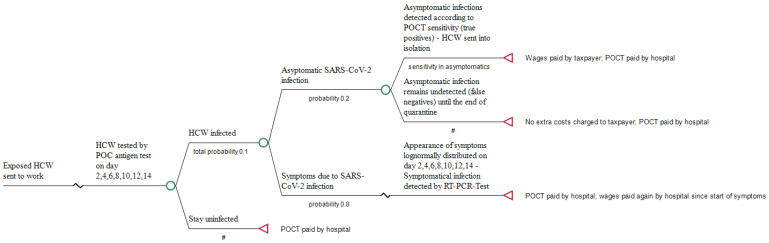
Pathway of sending exposed HCW to work (Scenario 2). #: Complementary probability (all probabilities of chance node’s branches to sum to 1.0).

**Table 1 ijerph-18-10767-t001:** Cost parameters used in the model.

Variables Category	Value (Mean Costs in EUR)	Reference
Costs of POCT antigen test (taking the Sofia SARS Antigen FIA^®^ as example)	12	[14]
Preanalytics of the POC antigen test and waiting for its result t (30 min)	14.88	Calculated from [26]
Wage costs per HCW and day	167.58	Calculated from [25]

**Table 2 ijerph-18-10767-t002:** Number of infected HCW under quarantine on different days.

Persons Newly Infected (No.)	d2	d4	d6	d8	d10	d12	d14	Sum after 14 Days
TP (S)	5.652	24.32	19.216	6.003	5.476	2.285	0.928	63.88
FN (S)	1.413	6.08	4.804	1.501	1.369	0.571	0.242	15.98
TP (NS)	1.024	4.408	3.483	1.088	0.993	0.414	0.177	11.587
FN (NS)	0.742	3.192	2.522	0.788	0.719	0.3	0.27	8.553
total	8.831	38	30.025	9.38	8.557	3.57	1.637	100

Legend to Table 1: Table 1 denotes the number of infected HCW in a hypothetical cohort of 1000 exposed HCW under quarantine on day 2, 4, 6, 8, 10, 12 and 14. d: day, TP: true positive, FN: False negative; S: symptomatic disease; NS: asymptomatic infection.

**Table 3 ijerph-18-10767-t003:** Costs of sending HCW into quarantine (per 1000 exposed HCW).

(a) Costs at the expense of the employer (hospital)	
Costs due to continuation of wage for all symptomatically diseased HCW, starting from the respective day of symptoms until the end of quarantine: *	
7.065 infected HCW on day 2 × EUR 167.58 × 12 remaining days	EUR 14,207.43
30.4 infected HCW on day 4 × EUR 167.58 × 10 remaining days	EUR 50,944.32
24.02 infected HCW on day 6 × EUR 167.58 × 8 remaining days	EUR 32,202.17
7.504 infected HCW on day 8 × EUR 167.58 × 6 remaining days	EUR 7545.12
6.845 infected HCW on day 10 × EUR 167.58 × 4 remaining days	EUR 4588.34
2.856 infected HCW on day 12 × EUR 167.58 × 2 remaining days	EUR 957.22
0.956 infected HCW on day 14 × EUR 167.58	EUR 196.07
Total cost for 1000 HCW sent into quarantine: EUR 110,640.67 (±20%: EUR 88,512.54 to EUR 132,768.80)	
Cost per exposed HCW sent into quarantine from the hospital´s perspective: EUR 110.64 (±20%: EUR 88.51 to EUR 132.77)	
(b) Costs at the expense of the taxpayer	
Reimbursement of wages to the employer (hospital) for all HCW sent into quarantine as long as no symptoms appear: 1000 HCW × EUR 167.58 × 14 days (EUR 2346,120; ±20%: 1876,896 to EUR 2815,344) minus EUR 110,604.67 (±20%: EUR 88,512.54 to EUR 132,768.80; see Table 3a)	
Total cost for 1000 HCW sent into quarantine: EUR 2235,479.33 (±20%: EUR 1744,127.20 to EUR 2726,831.46)	
Cost per exposed HCW sent into quarantine from the taxpayer´s perspective: EUR 2235.48 (±20%: EUR 1744.13 to EUR 2726.83)	

* The number of all HCW becoming symptomatically infected on the respective days 2, 4, 6, 8, 10, 12 and 14 according to the point estimates of Table 2 are multiplied by the average wage costs per day and the number of days remaining to the end of the 14-day quarantine period.

**Table 4 ijerph-18-10767-t004:** Costs of allowing HCW to work (per 1000 exposed HCW).

(a) Costs at the expense of the employer (hospital)	
Costs of Point-of-Care Testing [TP (S) plus FN (S) plus [TP (NS)]	
1000 (on day 2) × EUR 26.88 testing costs	EUR 26,880
991.911 [1000–8.089] (on day 4) × EUR 26.88 testing costs	EUR 26,662.57
957.103 [991.911–34.808] (on day 6) × EUR 26.88 testing costs	EUR 25,726.93
929.6 [957.103–27.503] (on day 8) × EUR 26.88 testing costs	EUR 24,987.65
921.008 [929.6–8.592] (on day 10) × EUR 26.88 testing costs	EUR 24,756.70
913.17 [921.008–7.838] (on day 12) × EUR 26.88 testing costs	EUR 24,546.01
909.9 [913.17–3.27](on day 14) × EUR 26.88 testing costs	EUR 24,458.11
Subtotal costs: EUR 178,017.97 (±20%: EUR 142,414.38 to EUR 213,621.56)	
2. Costs due to continuation of wage for all symptomatically diseased HCW [TP (S) and FN (S)], starting from the respective day of symptoms until the end of quarantine:	
7.065 infected HCW on day 2 × EUR 167.58 × 12 remaining days	EUR 14,207.43
30.4 infected HCW on day 4 × EUR 167.58 × 10 remaining days	EUR 50,944.32
24.02 infected HCW on day 6 × EUR 167.58 × 8 remaining days	EUR 32,202.17
7.504 infected HCW on day 8 × EUR 167.58 × 6 remaining days	EUR 7545.12
6.845 infected HCW on day 10 × EUR 167.58 × 4 remaining days	EUR 4588.34
2.856 infected HCW on day 12 × EUR 167.58 × 2 remaining days	EUR 957.22
0.956 infected HCW on day 14 × EUR 167.58	EUR 196.07
Subtotal costs: EUR 110,640.67 (±20%: EUR 88,512.54 to EUR 132,768.80)	
Total cost for 1000 HCW sent to work: EUR 288,658.64 (±20%: EUR 230,926.92 to EUR 346,390.36)	
Cost per exposed HCW sent to work from the hospital´s perspective: EUR 288.66 (±20%: EUR 230.93 to EUR 346.39)	
(b) Costs at the expense of the taxpayer	
Costs of sending asymptomatically infected HCW into isolation [TP (NS)]	
1.024 × EUR 167.58 (on day 2) × 12 remaining days	EUR 2059.22
4.408 × EUR 167.58 (on day 4) × 10 remaining days	EUR 7386.93
3.483 × EUR 167.58 (on day 6) × 8 remaining days	EUR 4669.45
1.088 × EUR 167.58 (on day 8) × 6 remaining days	EUR 1093.96
0.993 × EUR 167.58 (on day 10) × 4 remaining days	EUR 665.63
0.414 × EUR 167.58 (on day 12) × 2 remaining days	EUR 138.76
0.177 × EUR 167.58 (on day 14)	EUR 29.66
Total cost for 1000 HCW sent to work: EUR 16,043.61 (±20%: EUR 12,834.89 to EUR 19,252.33)	
Cost per exposed HCW sent to work from the taxpayer´s perspective: EUR 16.04 (±20%: 12.83 to EUR 19.25)	

## Data Availability

Not applicable.

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
