# Peer review of "Point-of-Care COVID-19 Antigen Testing in Exposed German Healthcare Workers—A Cost Model"

_ijerph, 2021, doi:10.3390/ijerph182010767_

Round 1

Reviewer 1 Report

The article explores the cost-benefit of implementing point-of-care COVID-19 antigen testing (POCT) for exposed healthcare workers (HCW) in Germany versus quarantine, through a mathematical model. The simulation adopts both the hospital and the taxpayer perspectives, that I find appropriate. The paper is important for the current international situation but probably in future too to manage optimally pandemic situations in terms of hospital staff level as well as outbreak control.

The article is well written and structured. I recommend this publication in International Journal of Environmental Research and Public Health with minor comments and suggestions which could perhaps improve the readers' understanding of certain points, according to authors’ approval. Just a point to consider: I am not expert for the epidemiological model and current available data on COVID-19. The model is based on a systematic review and meta-analysis of observational researches; and I trust the authors in the choice of their parameters.

Introduction : The introduction is well written and presents with a clearly manner the problematic and the two strategies:

  • Sample-and-stay strategy = periodic POCT with at least once every 48 hours together with wearing of a highly effective FFP2 mask
  • Quarantine during a period of at least 14-days for a HCW becoming contact person. The quarantine ends if the HCW does not have symptoms and a negative result of an antigen test.

Methods:

  • Who is considered a HCW? The mean gross annual learning could varied according to the type of hospital employee and I would suggest to define clearly who is consider.
  • It seems that the cost of RT-PCR is not presented ? Shouldn't it be taken into account in the scenarios?
  • If too much HCW is quarantined, could a possible additional cost of replacement HCW be considered from the hospital's perspective? While the assumption may not be relevant, it could be argued?
  • A decision-tree representation could be help the readers to understand all the link and probability between the two scenarios.

Results

  • Tables 2 and 3 are not easy to read for several reasons and I suggest improving its comprehension by: 1) putting it on a single sheet (or putting the column headers back on), 2) adding headings for all the columns, 3) adding thousand separators and every lines separating the columns (table 3).
  • Page 5 line 187 : delete a dot at the end of the sentence
  • Page 6 line 204 : I would suggest to add “€48 (scenario 1) minus €16.04 (scenario 2)” ?

Discussion

  • Do the authors have any idea or hypothesis as to the proportion of HCW who would adopt scenario 2 and who would volunteer to return to work with these tests? This seems to be an important factor in the implementation of scenario 2.
  • The only limit could be supported by an hypothesis about the degree of generalization in Germany of this model or really not ? Nevertheless, I think the model is understandable enough to be easily used in other contexts.

Author Response

  • Who is considered a HCW? The mean gross annual learning could varied according to the type of hospital employee and I would suggest to define clearly who is consider.

Reply: All hospital workers are addressed who are in direct contact with a patient including physicians, nurses and auxiliary services staff. We have clarified this definition in the Introduction.

  • It seems that the cost of RT-PCR is not presented? Shouldn't it be taken into account in the scenarios?

Reply: In contrast, the costs of RT-PCR incurred by the hospital are reimbursed according to the German Hospital Finance Act (Krankenhausfinanzierungsgesetz, KHG).16 Accordingly, RT-PCR testing does not appear as a cost factor in our model. We have now inserted this sentence in the main text (2.3. costs of testing and outcomes) and provided a reference.

  • If too much HCW is quarantined, could a possible additional cost of replacement HCW be considered from the hospital's perspective? While the assumption may not be relevant, it could be argued?

Reply: This is an interesting idea, thank you! Unfortunately, in Germany nurses and physicians working in acute-care hospitals are highly sought after, so replacements would come at a highly unreasonable cost. Hence the necessity to send exposed hospital staff to work instead of into quarantine. We have now inserted a sentence to that effect in the Introduction section to underline the pressure under which German hospitals operate and have provided a reference: “As the number of qualified nurses and doctors for inpatient care in Germany is insufficient, short-term replacements are more than unrealistic.”

  • A decision-tree representation could be help the readers to understand all the link and probability between the two scenarios.

Reply:  Thank you for this hint. For clarity, we have now inserted simplified pathways for each scenario.

Results

  • Tables 2 and 3 are not easy to read for several reasons and I suggest improving its comprehension by: 1) putting it on a single sheet (or putting the column headers back on), 2) adding headings for all the columns, 3) adding thousand separators and every lines separating the columns (table 3).

Reply: We have now split the tables and presented the columns separately, below one another.

  • Page 5 line 187 : delete a dot at the end of the sentence

Reply: Done

  • Page 6 line 204 : I would suggest to add “€48 (scenario 1) minus €16.04 (scenario 2)” ?

Reply: Thank you, done!

Discussion

  • Do the authors have any idea or hypothesis as to the proportion of HCW who would adopt scenario 2 and who would volunteer to return to work with these tests? This seems to be an important factor in the implementation of scenario 2.

Reply: It is a limitation of the model that, to date, there is no evidence as to how many HCW would in fact follow the offer to return to work due to missing data. We inserted a respective sentence in the discussion section.

  • The only limit could be supported by an hypothesis about the degree of generalization in Germany of this model or really not ? Nevertheless, I think the model is understandable enough to be easily used in other contexts. 

Reply: Please see our comments above.

Reviewer 2 Report

This is an interesting study. Here are my thoughts on this study

1) Will the authors consider this as an economic evaluation? if so, I would recommend them to follow the CHEERS guideline.

https://www.bmj.com/content/346/bmj.f1049

2) In my understanding, the health effects derived from your scenarios need to convert into cost otherwise it is just a cost analysis. in that sense, please change the title.

3) Are you considering the Human Capital approach to calculate the cost due to productivity loss? if that is the case, please write this and also discuss the implication of that (any type of overestimation)

4) what is your assumption on cost related to isolation, is just productivity loss cost enough to consider? what about the healthcare-related cost for those who are true positive. I think this is a big limitation and the results will be underestimated.

5) Even if the TP health workers do not have any health complications, there are costs related to quarantine. Shall you not consider this?

6) It would be good to perform several sensitivity analyses regarding your assumptions of the probability of being TP, FN on different days to check the robustness of your findings.

Author Response

1) Will the authors consider this as an economic evaluation? if so, I would recommend them to follow the CHEERS guideline.

https://www.bmj.com/content/346/bmj.f1049

Reply: Thank you, for your recommendation. Having taken a deeper look to the CHEERS guidelines, we note that these predominately refer to incremental cost-effectiveness analysis, hence not fully appropriate for our calculations. Please see also our comments on point 2).

2) In my understanding, the health effects derived from your scenarios need to convert into cost otherwise it is just a cost analysis. in that sense, please change the title.

Reply: You are right. Formally, our analysis is a cost-cost analysis where the costs of 2 different scenarios are compared separately. We are not able to convert health effects into cost. Therefore we replaced “cost-benefit analysis” as header of the study with “cost model”.

3) Are you considering the Human Capital approach to calculate the cost due to productivity loss? if that is the case, please write this and also discuss the implication of that (any type of overestimation)

Reply: Or intention was only to evaluate the cost saving due to lesser wage costs that have to be paid by the hospital or reimbursed by the taxpayer. We are not able to address the intrinsic human capital approach which goes beyond by placing a monetary value on loss of health, such as the lost value of economic productivity due to ill health, disability, or premature mortality. Consequently, we replaced the term “indirect costs” in the abstract by “wage costs”

4) what is your assumption on cost related to isolation, is just productivity loss cost enough to consider? what about the healthcare-related cost for those who are true positive. I think this is a big limitation and the results will be underestimated.

Reply: Thank you for this important hint: It was the intention of our cost model to address only the wage cost arising due to isolation. We did not calculate the healthcare costs arising in case of symptoms requiring treatment. Those costs will be paid separately by the Public Statutory insurances and therefore lie beyond the perspective of the hospital as employer. Accordingly, we have now inserted a respective sentence at the end of the Introduction: “We did not calculate the healthcare costs arising by symptomatically infected HCW requiring treatment: such costs have to be paid by the German Public Statutory insurances and thus lie beyond the perspective of the hospital as employer and that of the taxpayer.”

5) Even if the TP health workers do not have any health complications, there are costs related to quarantine. Shall you not consider this?

Reply: We were not able to assess additional costs arising through quarantine, e.g., due to food suppliers or other services. To date such data have never evaluated in Germany. We have inserted a respective sentence at the end of the Introduction.

6) It would be good to perform several sensitivity analyses regarding your assumptions of the probability of being TP, FN on different days to check the robustness of your findings.

Reply: Indeed this would be very reasonable, but for a sensitivity analysis distributions or ranges are required. However, as you can see in the Appendix, the probabilities of TP, FP, FN, and TN could only be derived as point estimators from a log-normal distribution. Therefore a sensitivity analysis demonstrating the robustness of our findings is not possible. We mentioned that fact as a limitation the Discussion.

Reviewer 3 Report

This paper represents a nice study. Yet, it is not entirely clear that there is a problem that needs to be addressed and that the proposed form of testing is indeed the best solution. The authors must convince readers that there is indeed a problem with/for hospital workers. See below more detailed comments:

  1. While hospital staffing shortages have been recurrently reported in the media, actual staffing shortages have been a more controversial topic than usually thought. According to a report by the University of Constance (1) and the Leibniz Institute (RWI) (2), German hospitals have never been overwhelmed during the COVID-19 crisis.
  2. 2 about the PCR test: “These assays have nearly perfect sensitivity and are therefore well suited as “gold standard” for the diagnosis of COVID-19.” This is a contestable statement. See, e.g., (3), which states that “Further evidence and independent validation of covid-19 tests are needed.” See also (4) and, especially, (5), which concludes: “that at a cycle threshold (ct) of 25, about 70% of samples remained positive in cell culture (i.e. were infectious); at a ct of 30, 20% of samples remained positive; at a ct of 35, 3% of samples remained positive; and at a ct above 35, no sample remained positive (infectious) in cell culture (see diagram) This means that if a person gets a “positive” PCR test result at a cycle threshold of 35 or higher (as applied in most US labs and many European labs), the chance that the person is infectious is less than 3%. The chance that the person received a “false positive” result is 97% or higher.”

Some statements the authors make are contestable:

  1. 2 line 86: “highly effective FFP2 mask”. This is also a controversial statement (see, e.g., 6). According to a study (7), medical masks still have a penetration of 44 %. Other studies question the effectiveness of masks to protect persons against COVID-19 infections altogether (8).
  2. 3: “the transmission of COVID-19 to HCW depends on their vaccination coverage”. There is evidence that vaccine coverage does not reduce transmission (9). Furthermore, some studies warn that vaccination could enhance the severity of the disease (10). Specifically, a study on the COVID-19 vaccine (11), using monkey trials, found that while the monkeys were protected from serious illness, it did not prevent them from becoming infected with Covid-19 in the first place.

Other questions/comments:

  1. To what extent do your results depend on the distributional assumptions, i.e., that that incubation period is indeed lognormally distributed?
  2. 3: “the hospital as employer … bears also the cost of testing.” This is supposed to be changed. To what extent are the results of your study still relevant?
  3. It would be best to have a citation to support your number of 80% of asymptomatic infections. Note that the theory that COVID-19 spreads asymptomatically is also contested based on empirical studies on a large scale (12).
  4. How did you come up with a number of €12 for the costs of rapid tests?
  5. Study (13) reports a similar conclusion to yours, stressing the importance of rapid testing. You may want to cite their working paper.

References

  1. https://coronavis.dbvis.de/de/overview/map/lockdown-live
  2. https://www.bundesgesundheitsministerium.de/fileadmin/Dateien/3_Downloads/C/Coronavirus/Analyse_Leistungen_Ausgleichszahlungen_2020_Corona-Krise.pdf
  3. BMJ2020;369:m1808; doi: https://doi.org/10.1136/bmj.m1808
  4. Jia, X., Xiao, L., & Liu, Y. (2021). False negative RT-PCR and false positive antibody tests–Concern and solutions in the diagnosis of COVID-19. Journal of Infection82(3), 414-451.
  5. Jaafar, R., Aherfi, S., Wurtz, N., Grimaldier, C., Van Hoang, T., Colson, P., ... & La Scola, B. (2021). Correlation between 3790 quantitative polymerase chain reaction–positives samples and positive cell cultures, including 1941 severe acute respiratory syndrome coronavirus 2 isolates. Clinical Infectious Diseases72(11), e921-e921.
  6. Feng, S., Shen, C., Xia, N., Song, W., Fan, M., & Cowling, B. J. (2020). Rational use of face masks in the COVID-19 pandemic. The Lancet Respiratory Medicine8(5), 434-436.
  7. MacIntyre, C. R., Seale, H., Dung, T. C., Hien, N. T., Nga, P. T., Chughtai, A. A., ... & Wang, Q. (2015). A cluster randomised trial of cloth masks compared with medical masks in healthcare workers. BMJ open5(4), e006577.
  8. Kappstein, I. (2020). Mund-Nasen-Schutz in der Öffentlichkeit: Keine Hinweise für eine Wirksamkeit. Krankenhaushygiene up2date15(03), 279-297.
  9. Van Beek, J., Veenhoven, R. H., Bruin, J. P., Van Boxtel, R. A., De Lange, M. M., Meijer, A., ... & Luytjes, W. (2017). Influenza-like illness incidence is not reduced by influenza vaccination in a cohort of older adults, despite effectively reducing laboratory-confirmed influenza virus infections. The Journal of infectious diseases216(4), 415-424.
  10. Haynes, B. F., Corey, L., Fernandes, P., Gilbert, P. B., Hotez, P. J., Rao, S., ... & Arvin, A. (2020). Prospects for a safe COVID-19 vaccine. Science translational medicine12(568).
  11. van Doremalen, N., Lambe, T., Spencer, A., Belij-Rammerstorfer, S., Purushotham, J. N., Port, J. R., ... & Munster, V. J. (2020). ChAdOx1 nCoV-19 vaccine prevents SARS-CoV-2 pneumonia in rhesus macaques. Nature586(7830), 578-582.
  12. Cao, S., Gan, Y., Wang, C., Bachmann, M., Wei, S., Gong, J., ... & Lu, Z. (2020). Post-lockdown SARS-CoV-2 nucleic acid screening in nearly ten million residents of Wuhan, China. Nature communications11(1), 1-7.
  13. Gabler, J., Raabe, T., Röhrl, K., & von Gaudecker, H. M. (2021). The Effectiveness of Strategies to Contain SARS-CoV-2: Testing, Vaccinations, and NPIs. arXiv preprint arXiv:2106.11129.

Author Response

This paper represents a nice study. Yet, it is not entirely clear that there is a problem that needs to be addressed and that the proposed form of testing is indeed the best solution. The authors must convince readers that there is indeed a problem with/for hospital workers. See below more detailed comments:

  1. While hospital staffing shortages have been recurrently reported in the media, actual staffing shortages have been a more controversial topic than usually thought. According to a report by the University of Constance (1) and the Leibniz Institute (RWI) (2), German hospitals have never been overwhelmed during the COVID-19 crisis.

Reply:  Both reports address the number of occupied hospital beds during corona crisis but neither cover actual shortages of staffing. In fact, such numbers are difficult to determine because reporting them is not required by law; they may be distorted because beds were often kept free when the incidence of COVID-19 was increasing in the past. We agree that in the current situation in Germany, major staff shortages have not yet been reported. Of course we are not able to predict shortages in the future, but we here provide an economic model should they occur. We have now clarified our intention by firstly deleting replacing the word “is” putting pressure  by “might” and , secondly,  by  inserting the following half-sentence: “Although in the current situation in Germany few actual staff shortages have yet been reported,…”

  1. 2 about the PCR test: “These assays have nearly perfect sensitivity and are therefore well suited as “gold standard” for the diagnosis of COVID-19.” This is a contestable statement. See, e.g., (3), which states that “Further evidence and independent validation of covid-19 tests are needed.” See also (4) and, especially, (5), which concludes: “that at a cycle threshold (ct) of 25, about 70% of samples remained positive in cell culture (i.e. were infectious); at a ct of 30, 20% of samples remained positive; at a ct of 35, 3% of samples remained positive; and at a ct above 35, no sample remained positive (infectious) in cell culture (see diagram) This means that if a person gets a “positive” PCR test result at a cycle threshold of 35 or higher (as applied in most US labs and many European labs), the chance that the person is infectious is less than 3%. The chance that the person received a “false positive” result is 97% or higher.”

Reply: You are right. As our study only addressed antigen-testing, not PCR-testing, we have deleted this sentence completely.

Some statements the authors make are contestable:

  1. 2 line 86: “highly effective FFP2 mask”. This is also a controversial statement (see, e.g., 6). According to a study (7), medical masks still have a penetration of 44 %. Other studies question the effectiveness of masks to protect persons against COVID-19 infections altogether (8).

Reply: Thank you for this statement. Our reference was a well-done meta-analysis of Chu et al. on the effectiveness of FFP2 masks in the hospital setting, where it can be trusted that these masks are worn properly.  The study of MacIntyre et al. (reference 7) was published in 2015 and does not refer to the effectiveness of FFP2 masks against COVID-19, Kappstein’s statement (reference 8) does not generally cast doubt upon the effectiveness of masks but states the opinion that in public (not in a hospital setting) a mouth and nose protection (not specifically a FFP2 mask) is not more efficient than keeping distance. Also Feng’s statement  (reference 6) is not an evaluation of the effectiveness of FFP2 masks in hospital but rather casts doubt on the effectiveness of universal face mask use in the community. All in all, we fell that there is no basis contradiction between the results of Chu´s meta-analysis cited by us and these studies. However, we fully agree that an effectiveness of only 67% in Chu´s analysis that we refer to can hardly be considered “highly” effective. Therefore we deleted the words “highly effective” in the Introduction.

  1. 3: “the transmission of COVID-19 to HCW depends on their vaccination coverage”. There is evidence that vaccine coverage does not reduce transmission (9). Furthermore, some studies warn that vaccination could enhance the severity of the disease (10). Specifically, a study on the COVID-19 vaccine (11), using monkey trials, found that while the monkeys were protected from serious illness, it did not prevent them from becoming infected with Covid-19 in the first place.

Reply: According to the official statements of the German Robert Koch Institute (https://www.rki.de/SharedDocs/FAQ/COVID-Impfen/FAQ_Transmission.html) the risk of transmission after vaccination is drastically reduced. However, as transmissions may still occur we changed the respective sentence. It now reads: “However, as transmissions may occur despite vaccination, transmission of COVID-19 to HCW depends not only on vaccination coverage, but also on the use of PPE in the investigated working units”.

Other questions/comments:

  1. To what extent do your results depend on the distributional assumptions, i.e., that that incubation period is indeed lognormally distributed?

Reply: As stated in the Appendix, the probabilities of FN, TP, FN and FP on the different days of testing are derived as point estimates from the lognormal distribution of infection probabilities during the 14 day-incubation period

  1. 3: “the hospital as employer … bears also the cost of testing.” This is supposed to be changed. To what extent are the results of your study still relevant?

Reply: Testing every two days as stated in the manuscript is a voluntary strategy offered by the employer which is not covered by the National testing regulation and consequently has to be paid for by the hospital itself. We do not have any information that the costs of testing will be covered by the responsible health authorities in the future.

  1. It would be best to have a citation to support your number of 80% of asymptomatic infections. Note that the theory that COVID-19 spreads asymptomatically is also contested based on empirical studies on a large scale (12).

Reply: We stated a sensitivity of 80% for detecting symptomatic infections by antigen testing (now reference 20) and a sensitivity of only 58% for asymptomatic infections (now reference 21). We have inserted two respective sentences in the Material and Methods Section

  1. How did you come up with a number of €12 for the costs of rapid tests?

Reply: This figure comes from one of our recently published study: Diel R, Nienhaus A. Point-of-care COVID-19 antigen testing in German emergency rooms - a cost-benefit analysis. Pulmonology. 2021 Jul 6:S2531-0437(21)00131-8. We have provided it as reference.

  1. Study (13) reports a similar conclusion to yours, stressing the importance of rapid testing. You may want to cite their working paper.

Reply: Thank you for mentioning the very interesting report (ref. 13) which we have read with great interest. We have cited the main conclusions of that work in the Discussion.  The sentences read: In a recent report of Gabler and coworkers it could be demonstrated that even in the period of increasing vaccination, rapid antigen testing had the largest effect on reducing the number of SARS-CoV-2 infections in Germany. Frequent and large-scale rapid testing should remain part of strategies to contain COVID-19.

  1. It would be good to perform several sensitivity analyses regarding your assumptions of the probability of being TP, FN on different days to check the robustness of your findings.

Reply: Indeed this would be very reasonable, but for a sensitivity analysis distributions or ranges are required. However, as you can see in the Appendix, the probabilities of TP, FP, FN, and TN could only be derived as point estimators from a log-normal distribution. Therefore a sensitivity analysis demonstrating the robustness of our findings is not possible. We mentioned that fact as a limitation the Discussion.

References

  1. https://coronavis.dbvis.de/de/overview/map/lockdown-live
  2. https://www.bundesgesundheitsministerium.de/fileadmin/Dateien/3_Downloads/C/Coronavirus/Analyse_Leistungen_Ausgleichszahlungen_2020_Corona-Krise.pdf
  3. BMJ2020;369:m1808; doi: https://doi.org/10.1136/bmj.m1808
  4. Jia, X., Xiao, L., & Liu, Y. (2021). False negative RT-PCR and false positive antibody tests–Concern and solutions in the diagnosis of COVID-19. Journal of Infection82(3), 414-451.
  5. Jaafar, R., Aherfi, S., Wurtz, N., Grimaldier, C., Van Hoang, T., Colson, P., ... & La Scola, B. (2021). Correlation between 3790 quantitative polymerase chain reaction–positives samples and positive cell cultures, including 1941 severe acute respiratory syndrome coronavirus 2 isolates. Clinical Infectious Diseases72(11), e921-e921.
  6. Feng, S., Shen, C., Xia, N., Song, W., Fan, M., & Cowling, B. J. (2020). Rational use of face masks in the COVID-19 pandemic. The Lancet Respiratory Medicine8(5), 434-436.
  7. MacIntyre, C. R., Seale, H., Dung, T. C., Hien, N. T., Nga, P. T., Chughtai, A. A., ... & Wang, Q. (2015). A cluster randomised trial of cloth masks compared with medical masks in healthcare workers. BMJ open5(4), e006577.
  8. Kappstein, I. (2020). Mund-Nasen-Schutz in der Öffentlichkeit: Keine Hinweise für eine Wirksamkeit. Krankenhaushygiene up2date15(03), 279-297.
  9. Van Beek, J., Veenhoven, R. H., Bruin, J. P., Van Boxtel, R. A., De Lange, M. M., Meijer, A., ... & Luytjes, W. (2017). Influenza-like illness incidence is not reduced by influenza vaccination in a cohort of older adults, despite effectively reducing laboratory-confirmed influenza virus infections. The Journal of infectious diseases216(4), 415-424.
  10. Haynes, B. F., Corey, L., Fernandes, P., Gilbert, P. B., Hotez, P. J., Rao, S., ... & Arvin, A. (2020). Prospects for a safe COVID-19 vaccine. Science translational medicine12(568).
  11. van Doremalen, N., Lambe, T., Spencer, A., Belij-Rammerstorfer, S., Purushotham, J. N., Port, J. R., ... & Munster, V. J. (2020). ChAdOx1 nCoV-19 vaccine prevents SARS-CoV-2 pneumonia in rhesus macaques. Nature586(7830), 578-582.
  12. Cao, S., Gan, Y., Wang, C., Bachmann, M., Wei, S., Gong, J., ... & Lu, Z. (2020). Post-lockdown SARS-CoV-2 nucleic acid screening in nearly ten million residents of Wuhan, China. Nature communications11(1), 1-7.
  13. Gabler, J., Raabe, T., Röhrl, K., & von Gaudecker, H. M. (2021). The Effectiveness of Strategies to Contain SARS-CoV-2: Testing, Vaccinations, and NPIs. arXiv preprint arXiv:2106.11129.

Reviewer 4 Report

This is a study that compares the cost of quarantine and no quarantine among HCWs from the employer's and tax payer's perspective but I have many concerns regarding this study.

  1. Is this a cost-benefit analysis (CBA)? CBA measures both costs and effects of interventions in monetary terms and usually involves placing a monetary value on health benefits, so what is the health benefit in monetary value in this study?
  2. The CHEERS checklist that has been widely has been widely adopted by major journals has not been adhered to. Can the authors use this checklist when reporting their findings please?
  3. Abstract:
    1. "saves €2219 per HCW in favor of the taxpayer..." This study is not comparing tax payer's with employer's perspective so why would it be "in favor of the tax payer"? Shouldn't it be in favor of either the intervention or comparator (quarantine vs no quarantine) that the authors are investigating?
  4. Methods:
    1. Can the authors state clearly in the articles what are the costs included in the employer's perspective and what are the costs included in the tax payer's perspective? In addition, can a table of unit cost with its references be provided please?
    2. What is the indirect cost since the abstract mentioned "indirect cost"?
    3. Is the time horizon of the model 15 days (day 0 to day 14)? Why wasn't a longer time horizon looked at?
    4. What is the discount rate used if relevant?
    5. What is the base year of the currency used?
    6. Where is the diagram of the model?
    7. What is the population included in the model?
    8. Given that tests to detect Covid-19 might not be 100% specific and sensitive, how would this cost the tax payer and employer if there is no quarantine compared to quarantine? What about the cost of infecting patients that the HCWs are taking care of when there is no quarantine and the test has false negatives? Would this not be of interest to the hospital as it further strains their resources?
    9. Has the model been validated; if so, how? If it is not validated, why not?
    10. "If employees decide to continue working, wear FFP2 masks and undergo an antigen test every other day at the workplace, the hospital as employer has not only to pay all wage costs for its HCW who continue working and those who are symptomatically infected..." Are the authors implying the employer would not have to pay the wage of their HCWs even if their employees choose not to continue working (i.e. paid sick leave)? Is paid sick leave excluded in this study since it will be the same from the employer's and tax payer's perspective?
    11. Description of statistical analysis is missing/ no statistical analysis in a cost-benefit analysis.
    12. Why wasn't uncertainty analyses conducted?
  5. Results:
    1. Is this the mean cost or median cost? Where is the standard deviation? What about its 95%CI?
    2. In Table 3, the scenario is "no quarantine" but there is a cost of "sending asymptomatically infected HCW into isolation"? Why is that so?
    3. Can a caption be included for all tables in the Appendix please?
    4. "All in all, when comparing the cost of the two scenarios, the net cost at the expense of the taxpayer is €2235.48 minus €16.04 that would be incurred if the second scenario came into effect, i.e. €2219.44 per exposed HCW. As the sick-leave costs to the hospital as employer is the same in both scenarios, the net cost to the hospital, following the second strategy, is limited to the costs of performing the antigen tests (€178.02 per exposed HCW)." Does that mean it is cost saving to not quarantine from the tax payer's perspective but it is cost saving to quarantine from the employer's perspective?

Author Response

Reviewer 4: This is a study that compares the cost of quarantine and no quarantine among HCWs from the employer's and tax payer's perspective but I have many concerns regarding this study.

  1. Is this a cost-benefit analysis (CBA)? CBA measures both costs and effects of interventions in monetary terms and usually involves placing a monetary value on health benefits, so what is the health benefit in monetary value in this study?

Reply: Formally, our analysis is a cost-cost analysis where the costs of 2 different scenarios are compared separately. We are not able to convert health effects into cost. Of primary importance for gaining the results of scenario 2 (no quarantine) are the epidemiological calculations of the probabilities of being infected at different times during the incubation period after being exposed to SARS-CoV 2. Therefore we replaced “cost-benefit analysis” by “cost model”.

  1. The CHEERS checklist that has been widely has been widely adopted by major journals has not been adhered to. Can the authors use this checklist when reporting their findings please?

Reply. Thank you, for your recommendation. Having a deeper look to the CHEERS guidelines these predominately refer to incremental cost-effectiveness analysis and thus are not fully appropriate for the calculations of our cost model. Please see also our statements on point 1).

Abstract:

    1. "saves €2219 per HCW in favor of the taxpayer..." This study is not comparing tax payer's with employer's perspective so why would it be "in favor of the tax payer"? Shouldn't it be in favor of either the intervention or comparator (quarantine vs no quarantine) that the authors are investigating?

Reply: Thank you, we absolutely agree. We have now rephrased the results of the two scenarios in the abstract and the main text. This study compares two scenarios, sending HCW into quarantine or not.

Methods:

    1. Can the authors state clearly in the articles what are the costs included in the employer's perspective and what are the costs included in the tax payer's perspective? In addition, can a table of unit cost with its references be provided please?

Reply: We have now improved the visibility of Tables 2 and 3 where the costs of performing scenario 1 and 2 are juxtaposed. The unit cost of POC testing has now been referenced, the other costs were been derived by statistical data that we already had referenced. The unit costs are now shown in an separate table..

  • What is the indirect cost since the abstract mentioned "indirect cost"?

Reply: The indirect costs are the wage costs. To avoid any misunderstanding, we have now replaced “indirect costs” by “wage costs”.

    1. Is the time horizon of the model 15 days (day 0 to day 14)? Why wasn't a longer time horizon looked at?

Reply: As assigned in para 2.1 and in Appendix, the time horizon is 14 days (starting with day 1 following exposure as day 0, the presumed day of exposure, will not be counted). As an incubation period is generally considered 14 days after exposure and exposed persons may leave quarantine at day 14 if they are tested-negative no costs arise beyond the period of 14 days.

    1. What is the discount rate used if relevant?

Reply: Thank you for this hint: Usually costs that have to be paid after 14 days do not have to be discounted. We have inserted a respective sentence in the text.

    1. What is the base year of the currency used?

Reply: The base year is 2021. We have now inserted this information in the text.

    1. Where is the diagram of the model? 

Reply:  Thank you for this hint. We have now inserted simplified pathways for both scenario to increase clarity.

    1. What is the population included in the model?

Reply: This study is no clinical trial. We stated in the text, all HCW in a German hospital which are exposed against an unprotected COVID -19 are addressed by law.

    1. Given that tests to detect Covid-19 might not be 100% specific and sensitive, how would this cost the tax payer and employer if there is no quarantine compared to quarantine? What about the cost of infecting patients that the HCWs are taking care of when there is no quarantine and the test has false negatives? Would this not be of interest to the hospital as it further strains their resources?

Reply: Ideed, as stated in the text and demonstrated in the Appendix the antigen tests are not 100% sensitive and specific but have a specificity of 98.9% and a sensitivity of 80% in symptomatic patients and 58% in asymptomatic patients. Regarding to your question, we wrote: “We assume that HCW who present COVID-like symptoms but are negative by POCT, as well as HCW who test positive by POCT, will be immediately re-tested by RT-PCR and that any SARS-CoV infections will be detected and/or confirmed, without further delay. Thus, in the second scenario, only 0.84% asymptomatic but infected HCW remain undetected and are not placed in quarantine”.

As simultaneous wearing of FFP2-masks is mandatory for those HCW who do not go into quarantine we further stated in the Discussion: “Of note, also the number of false-negative results in asymptomatically infected HCW infections during the 14-days incubation period is low (0.84 persons out of 100 HCW). Given the fact that wearing FFP2 masks reduces the risk of contagion by 67% [30], the risk of these subjects infecting other colleagues or patients by HCW left undetected by antigen testing can be considered tolerable (2.8 persons out of 1000 exposed HCW).” Thus we do not feel that this tiny number of infected, but undetected HCW may further strain the costs of the hospital as employer.

    1. Has the model been validated; if so, how? If it is not validated, why not?

Reply: This is a theoretical cost model based on statistical and epidemiological derivations. Of course, as we had stated at the end of the discussion, the model has to be validated in the future.

    1. "If employees decide to continue working, wear FFP2 masks and undergo an antigen test every other day at the workplace, the hospital as employer has not only to pay all wage costs for its HCW who continue working and those who are symptomatically infected..." Are the authors implying the employer would not have to pay the wage of their HCWs even if their employees choose not to continue working (i.e. paid sick leave)? Is paid sick leave excluded in this study since it will be the same from the employer's and tax payer's perspective?

Reply: Thank you for this question. Exactly, it was so stated in the text (2.3. Costs of testing and outcomes, first para). If the employees go to quarantine, the hospital has not to pay the wages for the quarantine period; the wages will be paid by the taxpayer instead. However, If the HCW develop symptoms and thus are sick, the hospital has to pay the wages again since appearance of the employees´ symptoms by law. We have now inserted the following sentence for further clarification: “That means, that during the 14-day period of quarantine the taxpayer has to bear the wage costs as long as no symptoms due to SARS-CoV-2 infection appear.”

    1. Description of statistical analysis is missing/ no statistical analysis in a cost-benefit analysis.

Reply: Sorry, but the statistical derivation of the point estimates is explained in detail in the Appendix. Other statistical analyses were not required.

    1. Why wasn't uncertainty analyses conducted?

Reply: Indeed this would be very reasonable, but for a sensitivity analysis distributions or ranges are required. However, as you can see in the Appendix, the probabilities of TP, FP, FN, and TN could only be derived as point estimators from a log-normal distribution. Therefore a sensitivity analysis demonstrating the robustness of our findings is not possible. We mentioned that fact as limitation the Discusssion.

  1. Results:
    1. Is this the mean cost or median cost? Where is the standard deviation? What about its 95%CI?

Reply: This calculation adds all attributable costs derived by fixed or pre-weighted costs. It does not take into consideration outcomes from clinical trials or different size samples. Therefore we always use mean costs. Confidence intervals are not provided, as their application to this non-probabilistic model would have been inappropriate. We inserted this sentence in the text for further clarification

    1. In Table 3, the scenario is "no quarantine" but there is a cost of "sending asymptomatically infected HCW into isolation"? Why is that so?

Reply: As stated in para 2. 3 (Incubation period and prevalence of infections in exposed HCW, lines 152 ff.), HCW who are tested positive but remain asymptomatic,  will be sent into isolation by the responsible public health department, and all wages, again, are covered by the tax-payer.

    1. Can a caption be included for all tables in the Appendix please?

Reply: Yes, course! Done as requested.

    1. "All in all, when comparing the cost of the two scenarios, the net cost at the expense of the taxpayer is €2235.48 minus €16.04 that would be incurred if the second scenario came into effect, i.e. €2219.44 per exposed HCW. As the sick-leave costs to the hospital as employer is the same in both scenarios, the net cost to the hospital, following the second strategy, is limited to the costs of performing the antigen tests (€178.02 per exposed HCW)." Does that mean it is cost saving to not quarantine from the tax payer's perspective but it is cost saving to quarantine from the employer's perspective?

Reply: Yes, exactly. That´s the results of the two scenarios. We have now also clarified these main messages in the Abstract.

Round 2

Reviewer 3 Report

The authors have satisfactorily addressed my concerns.

Author Response

Thank you very much for your kindest reply.

Reviewer 4 Report

I do not agree with the author's response of excluding sensitivity analysis; they could have conducted a best case-worst case scenario. Can the authors explain why this was not conducted please?

Author Response

We felt that, according to the significant gap between the taxpayer´s expenses following Scenario 1 and the costs to the employing hospital following Scenario 2, a reversal of the advantages of following Scenario 2 cannot be expected – with or without a sensitivity analysis. Nevertheless, following a conservative approach, we have now performed a sensitivity analysis by varying all costs by ± 20% and recalculated the costs of the two scenarios. As a result, the considerable burden of the taxpayer following Scenario 1, even if reduced by 20%, will by far not be outweighed by the increased costs at the expense of the hospital when following Scenario 2. We have inserted and discussed the results in the text. Once again, we would like to point out that further sensitivity analysis is limited by the fact that we cannot vary the log-normal distribution of the probabilities of becoming infected during the quarantine period.